# CNN–LSTM Neural Network for Identification of Pre-Cooked Pasta Products in Different Physical States Using Infrared Spectroscopy

**DOI:** 10.3390/s23104815

**Published:** 2023-05-17

**Authors:** Penghui Sun, Jiajia Wang, Zhilin Dong

**Affiliations:** 1School of Information Science and Engineering, Xinjiang University, Urumqi 830017, China; sunph@stu.xju.edu.cn (P.S.); dzl@stu.xdu.edu.cn (Z.D.); 2The Key Laboratory of Signal Detection and Processing, Xinjiang Uygur Autonomous Region, Xinjiang University, Urumqi 830017, China; 3Post-Doctoral Workstation of Xinjiang Xinjiang Uygur Autonomous Region Product Quality Supervision and Inspection Institute, Urumqi 830011, China

**Keywords:** infrared spectroscopy, pre-cooked pasta, deep learning, convolutional neural networks, LSTM

## Abstract

Infrared (IR) spectroscopy is nondestructive, fast, and straightforward. Recently, a growing number of pasta companies have been using IR spectroscopy combined with chemometrics to quickly determine sample parameters. However, fewer models have used deep learning models to classify cooked wheat food products and even fewer have used deep learning models to classify Italian pasta. To solve these problems, an improved CNN–LSTM neural network is proposed to identify pasta in different physical states (frozen vs. thawed) using IR spectroscopy. A one-dimensional convolutional neural network (1D-CNN) and long short-term memory (LSTM) were constructed to extract the local abstraction and sequence position information from the spectra, respectively. The results showed that the accuracy of the CNN–LSTM model reached 100% after using principal component analysis (PCA) on the Italian pasta spectral data in the thawed state and 99.44% after using PCA on the Italian pasta spectral data in the frozen form, verifying that the method has high analytical accuracy and generalization. Therefore, the CNN–LSTM neural network combined with IR spectroscopy helps to identify different pasta products.

## 1. Introduction

In the last few decades, there has been an increasing focus on the nutritional quality and convenience of food products. Pasta is a typical representative of the growing cooked wheat food market in the world due to its important nutritional properties and eating convenience [1]. In 2019, the world produced nearly 16 million tons of pasta, more than twice the 7 million tons produced 20 years ago. Italy is the largest producer of pasta (about 3.5 million tons), of which 60% of Italian pasta production is exported [2].

Pasta has a neutral taste and can be eaten with meat dishes or added to salads and desserts [3]. Consumers in the Chinese market can buy a wide variety of pasta. In addition, pasta contains a lot of energy and nutrients and is a good source of carbohydrates. A 100-g serving of pasta (cooked, unsalted) contains about 31 g of carbohydrates, 26.01 g of starch, 1.8 g of dietary fiber, 5.8 g of protein, and 0.93 g of lipids (fats) and provides about 158 calories [4]. Pasta has an average glycemic index of 52, with 64% being low-glycemic (Gi less than 55), depending on the type of pasta [5]. Pasta is easy to make and delicious. Since its spread, it has been loved by people all over the world. Even Chinese people who are not used to Western food can not resist the temptation of pasta. Of course, the reason for this, in addition to the deliciousness of pasta itself, is also related to its closeness to Chinese food tastes. Considering the ease of transportation and quality assurance, pasta products are transported using cold chain logistics. In this scenario, the need to ensure the quality of the product is a key concern for companies.

A series of chemical–physical analyses are necessary to ensure the quality of frozen pasta products and control the cold chain logistics. For manufacturing companies, the results of these analyses are not immediately available in many cases. Recently, pasta companies have started using infrared (IR) spectroscopy combined with chemometrics for the rapid determination of different parameters [6,7,8]. The Vis–SWIR spectrum was used for this experiment, which refers to the band between visible light and short-wave infrared. Similar to near-infrared spectroscopy, it can distinguish material information. In this paper, near-infrared spectroscopy and the Vis–SWIR spectrum are collectively referred to as IR spectroscopy.

IR spectroscopy is a commonly used spectroscopic technique, which is widely used for food quality testing [9], herb identification [10], and soil composition analysis [11]. Many machine learning methods have been proposed to extract the desired information from spectral data, including Naive Bayes (NB) [12], support vector machine (SVM) [13,14], random forest (RF) [15,16], K-nearest neighbor (KNN) [17], partial least squares (PLS) [18,19], etc. Since IR spectra introduce noise during the acquisition process, IR spectra contain not only the target information but also irrelevant information such as noise [20]. Therefore, preprocessing methods are usually used to eliminate the noise in the spectra; in addition, data downscaling is used to eliminate some of the overlapping information in the spectral data [21,22]. With the increase in the number of samples, the original pretreatment method is no longer applicable, and the accuracy and robustness of the established model decrease. Beyond this, the steps required for complex preprocessing and feature extraction are tedious. Learning features from raw data using deep learning methods has become the most popular research area.

In contrast to traditional machine learning methods, deep learning is an end-to-end approach [23]. End-to-end methods exhibit higher accuracy and better generalization performance than traditional methods. Recent studies have demonstrated the ability of one-dimensional convolutional neural networks (1D-CNN) to outperform traditional methods in qualitative analysis using IR spectroscopy. For example, 1D-CNN used infrared spectroscopy to identify coffee adulteration [24] and the qualitative analysis of blended fabrics [25]. The 1D-CNN proposed in this experiment used the Alexnet framework, and the Alexnet model was optimally adapted in this experiment. After changing the 2D convolutional layers to 1D convolutional layers, all pooling layers were removed, a Batch Normalization (BN) layer was added in front of the first three convolutional layers, and a fully connected layer was reduced. The BN layer was placed in front of the convolutional layers in an aligned sequence to speed up the Alexnet network training and prevent the model from overfitting. Since spectrographic data are in an ordered sequence, the absorbance values at different wavelength points contain positional information. Nevertheless, the CNN is not sensitive to feature locations, and recurrent neural networks (RNN) such as GRU, LSTM, etc., have been introduced into spectral analysis [26,27,28].

In the field of IR, most of the research on deep learning models is based on 1D-CNN, while the 1D-CNN combined with long short-term memory (LSTM) is relatively rare. Neural network models of 1D-CNN combined with LSTM are worthy models to be explored for analysis in IR. Neural network models of 1D-CNN combined with LSTM can be obtained from the hidden features of the learned samples at different locations of the data, thus better identifying different classes of samples [29]. Because spectral data arranged by wavelength or frequency amplitude behave similarly to time-frequency sequences, both the local abstract information of the spectra and the location information are crucial for the prediction of the target components. Therefore, there is a great need to develop an algorithm that can extract them from the local abstract and location information to improve the accuracy rate and robustness of the developed model.

In this study, this paper proposes a CNN–LSTM neural network to identify pasta product types in different physical states using IR spectra. In this paper, two learners, 1D-CNN and LSTM, are constructed to extract the local abstract information and the sequence position information from the spectra, respectively. Specifically, the data features inside the spectrum learned by the CNN are input into the LSTM, and the LSTM is used to extract the position information in the spectral data. To further highlight the superiority of the CNN–LSTM approach suggested in this research, the CNN–LSTM is compared with six other common calibration models, including the CNN, LSTM, SVM, DT, NB, and K-nearest neighbor (KNN) using an experimental pre-cooked pasta spectral dataset. This study focuses on the construction of an identification model for Italian pasta products to provide a theoretical basis for the rapid detection of pasta products. The main three contributions of this paper are as follows:A novel deep-learning model combined with IR spectroscopy is proposed for the identification of Italian pasta products.Compared with the CNN, LSTM, SVM, DT, plain Bayesian (NB), and K-nearest neighbor (KNN), our model achieves the highest performance.This paper analyzes the classification performance of deep learning in the thawed state outperforming the frozen state.

The rest of the paper is arranged as follows. Section 2 describes the CNN–LSTM neural network model proposed in this paper in detail, Section 3 gives the experimental results, and Section 4 summarizes the paper.

## 2. Materials and Methods

The flowchart of the study for this experiment is shown in Figure 1. The aim of this experiment is to normalize the raw IR spectra of pasta or extract the spectral features using principal component analysis (PCA), process the spectral data, and input them into traditional machine learning and deep learning models for analysis.

### 2.1. Datasets and Sample Partition

A portable spectrometer system and spectral acquisition was used in this experiment; a Vis–SWIR (Visible–Short Wave Infrared) portable ASD FieldSpec 4^®^ Standard–Res spectrophotometer was used to collect reflectance spectra.

The spectroradiometer operates in the 350 and 2500 nm wavelength range and is characterized by three detectors with different spectral resolutions: 3 nm at 700 nm and 10 nm at 1400/2100 nm. Controlled by a portable PC, the instrument consists of a detector housing and a fiber optic cable connected to a touch probe to perform reflectivity measurements.

The spectrophotometer has three different holographic diffraction gratings, each coupled to a detector. In the case of detectors, an order-separation filter covers each detector for rejection of second and higher order light.

The pasta samples analyzed in this study were provided by Gelit SpA, Cisterna di Latina (Lazio, Italy). The samples analyzed included two types of precooked pasta products with two different degrees of salting: 6 samples of Pennette72 and 6 samples of Mezze Penne. We placed approximately 200 g of each sample in separate plastic pans. The reflectance spectra were then collected in two stages: freezing (just out of the dry ice freezer) and thawing (after standing at 26 °C for about two hours).

The Vis–SWIR data were collected on 6 Pennette72 samples and 6 Mezze Penne samples with different salt contents under two physical conditions (frozen vs. thawed) with the help of contact probes. Fifty Vis–SWIR spectra were collected from each sample at each measurement time, yielding 1200 raw spectra [30].

After the systematic sampling of the dataset, it was divided into a training set and a testing set. The reasonable division of the training set and testing set is all-important for deep learning modeling. In order to verify the reliability of the CNN–LSTM model proposed in this experiment and retain as much prediction data as possible, the data were divided into a training set and a testing set at a ratio of 7:3.

Table 1 outlines the description of the dataset. The physical state of the pasta contained two conditions, frozen and thawed, the amount of data in each form was 600, and the number of features included in each spectrum was 2151.

### 2.2. Spectra Standardization and PCA

Standardization helps to accelerate the calculations in the algorithm. Standardization was performed using StandardScaler() [31] for all the divided datasets. The formula given below was applied for the transformation.
(1)$$S=\frac{x-\mu }{\sigma },$$
where S is the transformed value of the feature, *x* is the original value, *μ* is the mean, and *σ* is the standard deviation.

PCA is a classical linear dimensionality reduction method that preserves the main features and eliminates some information that overlaps the data [32]. It enables the linear combination of multiple independent variables according to the maximum variance principle and replaces the original variables with a small number of synthetic variables [33]. The linear model in the following equation represents the data matrix X:(2)$$X=TP^{T}+E,$$
where T is the fraction matrix $\;\left(n\;\times \;k\right)$, P is the load matrix $\left(m\times k\right)$, and E is the error matrix.

### 2.3. Hybrid 1D-CNN and LSTM Neural Network Model Development

The combination of CNNs and recurrent neural networks (RNNs) has had several research applications and has recently yielded significant results in the following areas: in the field of natural language processing, such as sentiment analysis and speech recognition [34,35]; in real-time data prediction, such as stock prediction [36,37], human energy consumption [38], and air pollution prediction [39,40]; and in computer vision, for modal frequency detection in computer vision [41]. The CNN reduces parameters through weight sharing to improve the model learning efficiency. In addition, the LSTM is a recurrent network model which solves the long-standing gradient explosion and gradient disappearance problems in the RNN [37].

The CNN–LSTM network framework is shown in Figure 2. In this study, two learners based on a 1D-CNN and LSTM were built. The two learners were constructed to extract the local abstraction and sequence position information from the spectra, severally. Specifically, the data features inside the spectrum learned by the CNN were input into the LSTM, and the LSTM was used to extract the position information in the spectral data. The CNN–LSTM model can fully extract the inherent characteristics of the data, and it also enables the model to have a certain ability to mine medium and long sequence data information. It can learn the hidden features of samples and obtain this information to better identify samples of different categories. In this paper, the model was further optimized to enhance its performance for infrared spectroscopy analysis and identification. The hidden layer size of the LSTM was 10, and the dropout parameter was 0.3.

#### 2.3.1. 1D-CNN

The 1D-CNN constructed in this paper utilized the Alexnet framework, and the structure of the 1D-CNN model is shown in Figure 3. Alexnet is a classical deep learning model that contains an 8-layer transform with 5 convolutional and 2 fully connected layers and a fully connected output layer. Alexnet adds the Rectified Linear Unit (ReLU) activation function after each convolutional layer, which makes Alexnet use a random discard technique to selectively ignore neurons in the fully connected layer during training to avoid overfitting the model [42].

The Alexnet model was optimally adapted for this experiment to identify the IR spectral data better. After changing the two-dimensional convolutional layers to one-dimensional convolutional layers, all the pooling layers were removed. A BN layer was added before the first three convolutional layers, and a fully connected layer was reduced. The arrangement sequence of the BN layers placed before the convolutional layers can speed up the Alexnet network training and prevent model overfitting.

When using the CNN to model one-dimensional IR spectral data, the more filters, the more redundant information will be generated, which will slow down the running speed, while fewer filters will not be able to fully extract the features; so, the appropriate number of filters can make the model more efficient in analyzing the one-dimensional spectral features and fully extract the features. The optimized and adjusted model parameters are shown in Table 2. In the 1D-CNN model, the batch size was 32, the optimizer Adam was selected, the learning rate (Ir) was set to 0.001, and the dropout was 0.5.

#### 2.3.2. LSTM

An LSTM is a recurrent neural network model that solves the long-standing gradient explosion and gradient disappearance problems of RNN models [37]. The LSTM units are shown in Figure 4, and the work of each LSTM unit in the architecture can be described as follows. For any input $x_{t}$ of spectral data, the LSTM model generates a hidden activation $h_{t}$, expressed by Equation (5). In the next step, each piece of spectral data is input and further utilized for prediction. The LSTM model defines the transformation relation $h_{t}$ of the hidden representation through the LSTM unit, which accepts the input $x_{t}$ in the current state and the acquired information $h_{t-1}$. Thus, when our LSTM network inputs spectral data, it processes the data and passes the inherited information to the next step.

Each LSTM cell contains a memory cell $\;C_{t}$, calculated using Equation (4). Each LSTM generates a new candidate ${\tilde{C}}_{t}$ to replace $C_{t}$ as a placeholder, and the ${\tilde{C}}_{t}$, as a memory, helps the hidden unit retain past spectral information, calculated using Equation (3). The memory unit is the most critical structure in the LSTM to avoid gradient disappearance and gradient explosion.
(3)$${\tilde{C}}_{t}=tanh\left(W_{c}\cdot \left[h_{t-1},x_{t}\right]+b_{c}\right)$$
(4)$$C_{t}=f_{t}*C_{t-1}+I_{t}*{\tilde{C}}_{t}$$
(5)$$h_{t}=o_{t}*\tan h\left(C_{t}\right)$$

Then, the LSTM generates an oblivion gate $f_{t}$, an update gate $i_{t}$, and an output gate $o_{t}$.

Oblivion Gate:(6)$$\;f_{t}=\sigma \left(W_{f}\cdot \left[h_{t-1},x_{t}\right]+b_{f}\right)$$
Update Gate:(7)$$i_{t}=\sigma \left(W_{i}\cdot \left[h_{t-1},x_{t}\right]+b_{i}\right)\;$$
Output Gate:(8)$$\;o_{t}=\sigma \left(W_{o}\cdot \left[h_{t-1},x_{t}\right]+b_{o}\right)\;$$
Among them, $W_{c}$, $W_{f}$, $W_{i}$, and $W_{o}\;$ are the weight parameters, and $b_{c}$, $\;b_{f}$, $b_{i}$, and $b_{o}$ are the deviation parameters. $C_{t}$ is generated by combining $C_{t-1}$, $h_{t-1}$, the input $h_{t-1}\;$ is the hidden state at the previous moment, and $h_{t}$ is the output at the current moment.

### 2.4. Model Evaluation Indexes

After training a model with the training set, we utilized the model to predict data in the testing set. *Accuracy*, *Precision*, *Recall*, and the *F*1-score are commonly utilized indicators in binary classification, and the formulas are as follows:(9)$$Accuracy=\left(\frac{TP+TN}{TP+FP+TN+FN}\right)\times 100\%$$
(10)$$Precision=\left(\frac{TP}{TP+FP}\right)\times 100\%$$
(11)$$Recall=\left(\frac{TP}{TN+FN}\right)\times 100\%$$
(12)$$F1=\frac{2•Precision•Recall}{Precision+Recall}.$$

In this experiment, the Pennette72 was labeled “1” and the Mezze Penne product was labeled “0”. In the above formulas, True Positive (*TP*) represents the number of samples with the label “1” for which the number of samples with label “1” was predicted. False positive (*FP*) represents the number of samples for which the label “0” was predicted as label “1”, and true negative (*TN*) represents the number of samples for which the label “0” was predicted as label “0”. False negative (*FN*) represents the number of samples where the label “1” was predicted as label “0”.

## 3. Results and Discussion

### 3.1. Subsection

As shown in Figure 5a, the spectra of “frozen pasta” showed higher reflectance values than that of “thawed pasta” in the following spectral ranges: 350–1450 nm, 1600–1850 nm, and 2100–2400 nm. The average spectra of the frozen pasta products showed more significant absorbance around 1450–1550 nm and 1900–2100 nm. As shown in Figure 5b, for the thawed pasta in the range 1300–2500 nm, the pasta product Pennette72 showed higher reflectance values than Mezze Penne, while the reflectance values of both products in the frozen state were almost identical.

### 3.2. Spectral Analysis Result Analysis

The CNN was compared with the KNN, DT, SVM, and NB traditional machine learning methods. In this paper, 100 features of spectral data were extracted using PCA and ranked in descending order from the contribution values. The code to build and optimize the model in this paper was run in Tensorflow 2.2.0 and the Keras 2.3.1 environment. The operating system was Windows 10 with an AMD Ryzen 7 5800H, Radeon Graphics 3.20 GHz processor, and 16.0 GB RAM.

The results presented in Table 3 show that the KNN, DT, NB, and SVM utilized the data after standardization, and the highest accuracy rate was 94.39% in the frozen state of the NB. The performance of the NB after using PCA was also very good, reaching the best accuracy rate of the model in both physical states, reaching 94.5% in the frozen state and 98.67% in the thawed state. Moreover, for all the machine learning methods in Table 3, the accuracy rate was improved after using standardization and PCA data processing, which shows that using preprocessing when using machine learning modeling can significantly improve the model performance. In the analysis of the raw data, the CNN model was better than all the traditional machine learning models, which indicates that the CNN end-to-end training advantage is undeniable, and it is better than KNN and SVM even after using data processing against machine learning. Furthermore, in Table 3, after the raw, standardization, and PCA processing, the CNN with one fully connected layer and one fully connected output layer achieved the highest accuracy compared with the traditional machine learning, which indicates that using a fully connected layer and a fully connected output layer of CNN is feasible. While the pasta accuracy of 85.41% in the frozen state was better than the 80% in the thawed state without treatment, the 96.11% in the frozen state with standardization was lower than the 97.78% in the thawed state and the 98.33% in the frozen state after processing with PCA, which was less than the 100% of the thawed state. This shows that the processing of data has different effects on pasta products in different physical states, but the use of preprocessing has a positive effect on the identification of pasta products. Using Alexnet to classify pasta products did not identify the pasta products with the original data. The data after standardization and PCA processing are shown in Table 3. It was only better than the CNN in the standardization frozen state. Alexnet combined with LSTM to become Alexnet–LSTM, the Alexnet–LSTM model was as poor as Alexnet in recognizing the unprocessed spectral data, and the accuracy rate under standardization processing in the frozen state was lower than Alexnet, while the performance was improved in other cases. The CNN–LSTM had an accuracy rate of 84.58% in the unprocessed frozen state, which was slightly lower than CNN’s 85.41%, and the accuracy rate was the best in the other cases. Compared with the Alexnet–LSTM, the CNN–LSTM was better in any state, which shows that the CNN–LSTM model is feasible.

As shown in Figure 6, when tuning the dropout and the learning rate of the CNN, the performance of the CNN was the best when the dropout was 0.5 and the Ir was 0.001. In Figure 7, when adjusting the dropout and layer size of the LSTM, it was found that when the dropout was 0.3 and the layer size was 10, the performance of the LSTM was optimal.

The performance of the CNN, LSTM, and CNN–LSTM models on the test set after standardization processing is shown in Table 4. The results presented in Table 4 show that the CNN–LSTM model was significantly improved compared with the CNN and LSTM. Among them, in the frozen state, the accuracy was 3.33% higher than that of the CNN and 11.11% higher than that of the LSTM; in the thawed state, it was 1.66% higher than that of the CNN and 6.66% higher than that of the LSTM. This shows that it is feasible to combine the CNN with the LSTM to improve the performance of the model. The CNN–LSTM showed the best accuracy in both physical states.

Since the number of features of the spectral data in this experiment reached 2151, this experiment further analyzed the classification effect after using PCA feature extraction, and the results are shown in Table 4. For the spectral data after feature extraction, both the CNN–LSTM and the CNN in the thawed state achieved 100% accuracy; the best model, the CNN–LSTM, in the frozen state had an accuracy rate of 99.44%.

The results presented in Table 4 show that both the CNN and CNN–LSTM achieved 100% performance after thawing after using the PCA, but the accuracy of the CNN–LSTM in the frozen state reached 99.44%, which was better than the CNN’s 98.33%. From the results of Table 4, the CNN–LSTM was better than CNN, and the CNN was better than LSTM. Moreover, in general, it is more accurate in identifying pasta in a thawed state than in a frozen state.

## 4. Conclusions

In this paper, a CNN–LSTM neural network was proposed for the accurate detection of pasta products using IR spectroscopy. The proposed model can extract both the spectral local abstraction information and the location information. The feasibility, portability, and generalization capabilities were verified using the frozen and thawed states of the pasta product dataset. The experimental results show that the pasta products in the thawed state are more suitable for classification when modeling based on a deep learning model.

To further show the superiority of the CNN–LSTM method proposed in this paper, other methods were compared with the method proposed in this paper. The results showed that the method achieved the highest accuracy in both pasta product datasets, further validating that the method has higher accuracy and better generalization for analysis. Bidirectional Long Short-Term Memory (BiLSTM) can effectively capture the dependence before and after the timing signal. The establishment of the CNN and the BiLSTM can extract spectral information from infrared spectra more effectively. Therefore, combining the CNN and BiLSTM for spectral recognition is a candidate direction of future research.

## Figures and Tables

**Figure 1 sensors-23-04815-f001:**
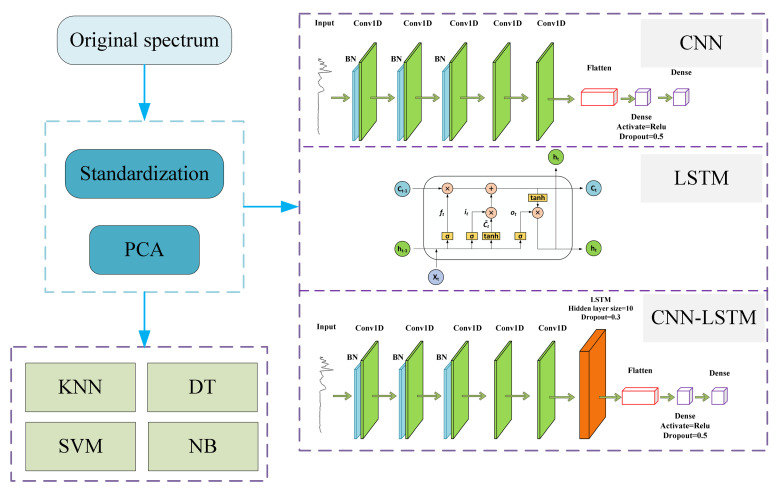
Flow chart of the study.

**Figure 2 sensors-23-04815-f002:**
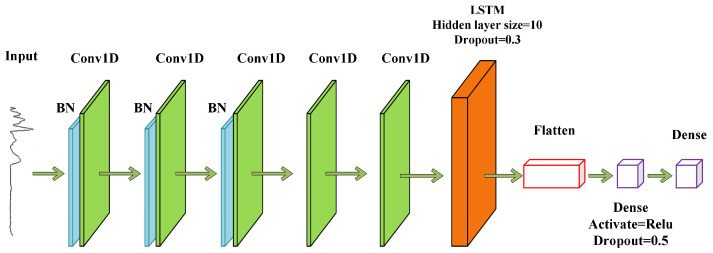
Network structure diagram of the adjusted CNN–LSTM.

**Figure 3 sensors-23-04815-f003:**
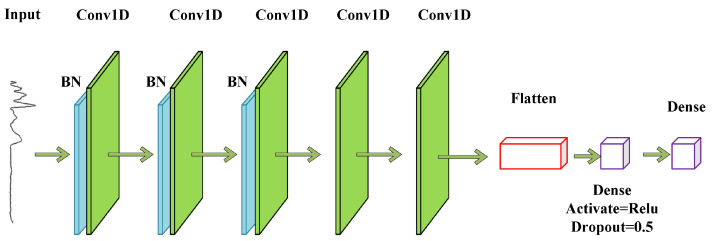
Network structure diagram of the 1D-CNN.

**Figure 4 sensors-23-04815-f004:**
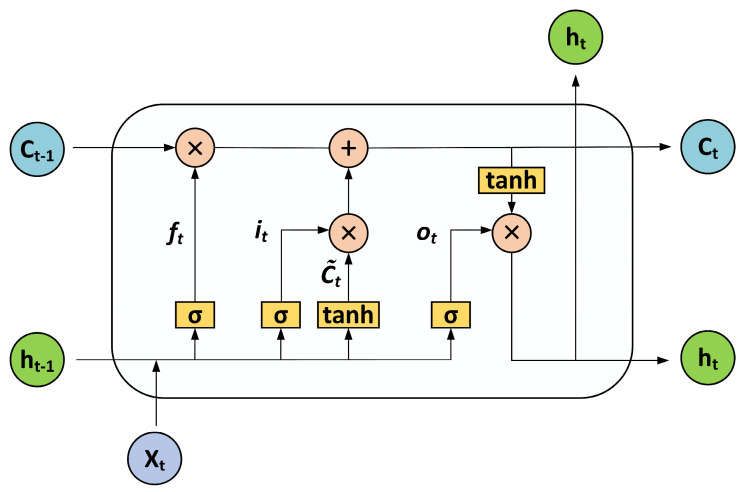
Network structure diagram of the LSTM.

**Figure 5 sensors-23-04815-f005:**
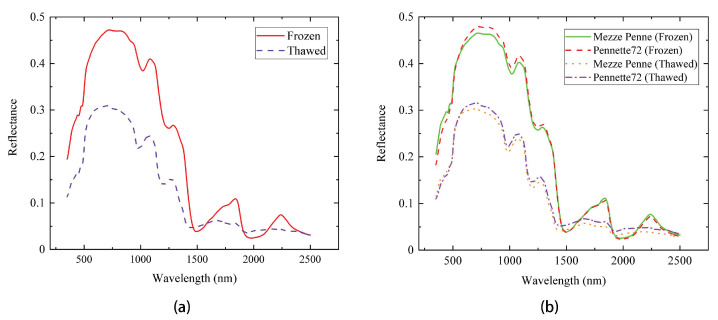
(**a**) IR spectra of pasta in the frozen and thawed states; (**b**) IR spectra of different brands of pasta in the frozen and thawed states.

**Figure 6 sensors-23-04815-f006:**
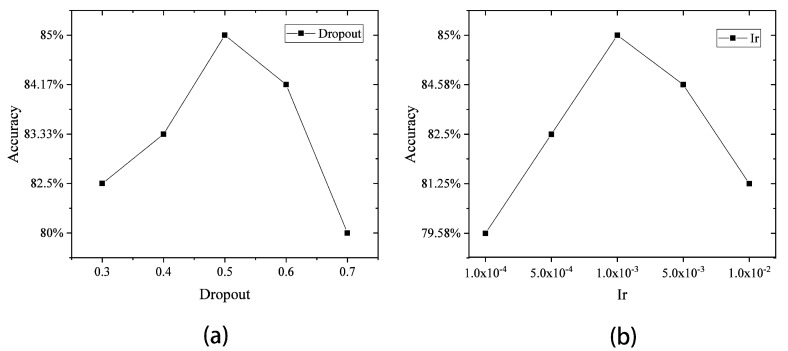
(**a**) Accuracy rate of the CNN with different dropouts; (**b**) accuracy rate of the CNN under different Ir.

**Figure 7 sensors-23-04815-f007:**
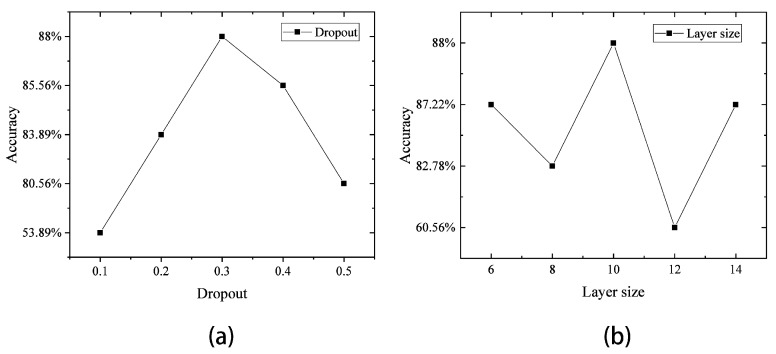
(**a**) Accuracy rate of the LSTM with different dropouts; (**b**) accuracy rate of the LSTM under different layer sizes.

**Table 1 sensors-23-04815-t001:** The characteristics of the pasta product.

Physical Condition	Training Samples	Testing Samples	Variables	Classes
Frozen pasta	420	180	2151	2
Thawed pasta	420	180	2151	2

**Table 2 sensors-23-04815-t002:** Model parameters of the CNN.

Operator	Filters	Kernel Size	Strides
Conv1D	64	11	4
	64	5	1
	92	3	1
	92	3	1
	64	3	1

**Table 3 sensors-23-04815-t003:** Accuracy of the deep learning and traditional machine learning models in the frozen and thawed state models.

Model	Physical Condition	Raw	Standardization	PCA
KNN	Frozen	72.67%	80.44%	72.67%
	Thawed	75.17%	75.72%	75.17%
DT	Frozen	76.39%	85.17%	86.39%
	Thawed	76.50%	76.61%	88.56%
NB	Frozen	52.94%	94.39%	94.5%
	Thawed	69.44%	69.44%	98.67%
SVM	Frozen	70.28%	83.50%	80.78%
	Thawed	74.44%	81.22%	77.78%
Alexnet	Frozen	-	99.17%	95.56%
	Thawed	-	96.67%	97.77%
CNN	Frozen	85.41%	96.11%	98.33%
	Thawed	80%	97.78%	100%
Alexnet–LSTM	Frozen	-	98.75%	96.11%
	Thawed	-	97.22%	98.82%
CNN–LSTM	Frozen	84.58%	99.44%	99.44%
	Thawed	80%	99.44%	100%

**Table 4 sensors-23-04815-t004:** Performance of deep learning in the frozen and thawed states.

Data Processing	Model	Physical Condition	*ACC*	*Precision*	*Recall*	*F*1-Score
Standardization	CNN	Frozen	96.11%	96.21%	96.05%	96.10%
		Thawed	97.78%	97.94%	97.70%	97.77%
	LSTM	Frozen	88.33%	88.33%	88.38%	88.33%
		Thawed	92.78%	93.27%	92.97%	92.77%
	CNN–LSTM	Frozen	99.44%	99.43%	99.46%	99.44%
		Thawed	99.44%	99.47%	99.43%	99.44%
PCA	CNN	Frozen	98.33%	98.33%	98.39%	98.33%
		Thawed	100%	100%	100%	100%
	LSTM	Frozen	96.11%	96.10%	96.13%	96.11%
		Thawed	91.06%	91.42%	91.27%	91.59%
	CNN–LSTM	Frozen	99.44%	99.47%	99.43%	99.44%
		Thawed	100%	100%	100%	100%

## Data Availability

Data used in this study may be obtained at [https://doi.org/10.17632/yhyzmp8rtb.2, accessed on 5 April 2023].

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
