# Peer review of "CNN–LSTM Neural Network for Identification of Pre-Cooked Pasta Products in Different Physical States Using Infrared Spectroscopy"

_sensors, 2023, doi:10.3390/s23104815_

Round 1

Reviewer 1 Report

1. In Section 3.2, the authors showed that CNN performs better than KNN, DT, NB, and SVM.  However, this is difficult to consider as a new contribution because many previous works already showed that CNN deep learning is superior to traditional machine learning (e.g. SVM, KNN).

2. The CNN-LSTM combination method has already been published in many previous papers, and it has already shown superior performance compared to the existing CNN and LSTM alone methods. Thus, it is difficult to consider it as a new contribution.

3. Regarding the dataset for proposed algorithm, the experimental contribution is also low because the authors cited the data from Reference [30] rather than the data obtained experimentally in their own way.

4. In addition, since the existing dataset is simply adopted, it is difficult for readers to determine which IR equipment and how to acquire data to apply the deep learning algorithm of the authors.

5. The authors shall provide product information and actual images(including IR images) for Frozen Pasta and Thawed Pasta in detail.

6. The authors mentioned Vis-SWIR data, but it is considered to be a different equipment than the general IR camera mentioned in the title and section 1. Authors shall provide further clarification and redefinition for this.

7. Line 171, the authors need to provide rationale for setting Layer size to 10 and Dropout to 0.3.

8. Line 194, the authors need to provide rationale for setting the learning rate to 0.001 and the dropout to 0.5.

Author Response

Dear Reviewers:

Thanks to the reviewers for your patient review and important suggestions for our article. We have improved the parts that you mentioned. We have highlighted our manuscript changes through the use of red font in the revised manuscript.

The revision was carried out as a result of the comments of the editor and the reviewers.

Yours sincerely,

Name: Jiajia Wang

E-mail: wjjxj@xju.edu.cn

Reviewer 2 Report

The paper deals with the CNN-LSTM Neural Network for Identification of Pre-Cooked 

Pasta Products in Different Physical States Using Infrared Spectroscopy. 

According to the reviewer, the paper is worth publishing at Sensors Journal, 

but corrections are needed and then the paper can be accepted for publication in the journal.

While the authors have made considerable research effort, 

the presentation of the paper must be improved. 

Additionally make the following corrections to the manuscript:

Comment 1

Line 16

is proposesd

Extended text editing.

Comment 2

Lines 29 - 34

The authors must format according to the journal's instructions (full text alignment).

Comment 3

Lines 98, 169, 208 and 216

It is not so good to use the word "we".

The authors should repfrace. 

Comment 4

Line 124

the help of contact probes.

The authors shourd give more details.

Comment 5

Line 132

The authors should consider if they insert a Table with a typical measurement data (from [30]).

Comment 6

Line 171

The hidden layer size of LSTM is 10, and the dropout parameter is 0.3.

The authors must explain how these values occurs. 

Comment 7

Line 178

The authors should explain the "Relu" (ReLU: Rectified Linear Unit?).

Comment 8

Lines 193 - 194

The authors must explain how these values occurs. 

Comment 9

Figure 4

thah?

The authors must explain (and check).

If it must change, the authors must change and the Figure 1.

Comment 10

Line 226

combining??−1

The authors should replace (insert a space):

combining ??−1

Comment 11

Line 245

It's not so good to start the subsection at the bottom of the page without using text.

Comment 12

Figure 5

(a)IR spectra

The authors should replace

(a) IR spectra

state;(b) IR spectra

The authors should replace

state; (b) IR spectra

Wavelength(nm)

The authors should replace

Wavelength (nm)

Mezze Penne(Frozen)

Pennette 72(Frozen)

Mezze Penne(Thawed)

Pennette 72(Thawed)

The authors should replace

Mezze Penne (Frozen)

Pennette 72 (Frozen)

Mezze Penne (Thawed)

Pennette 72 (Thawed)

Comment 13

Table 3

The authors must explain:

Model physical 

      condition  Raw         Standardization   PCA

CNN   Frozen     85.41%      96.11%            98.33%

CNN   Thawed     80%         97.78%            100%

While Raw is decreased for Thawed (85.41% > 80%), the Standardization and the PCA is increased (96.11% < 97.78% and 98.33% < 100%).

Comment 14

The authors should delete the Lines 313 - 315, 329 - 338 and 340 - 343.

Comment 15

Line 320

Validatio

Extended text editing.

Comment 16

References

The authors must format the References according to the journal's instructions

References should be described as follows, depending on the type of work:

Journal Articles:

1. Author 1, A.B.; Author 2, C.D. Title of the article. Abbreviated Journal Name Year, Volume, page range.

Author Response

(The authors gave the same response as above.)

Reviewer 3 Report

This manuscript proposed a novel hybridised method for identifying pre-cooked pasta products in different physical states based on infrared spectroscopy, where CNN and LSTM were employed for the task of interest. The experiment was conducted to validate the performance of proposed method, via comparing it with other models. The results show that accuracy of the CNN-LSTM model reached 100% after using principal component analysis, demonstrating excellent generalisation capacity. Overall, the topic of this research is interesting, and the manuscript was well organised and written. The detailed comments are summarised as follows.

1.       The main innovation and contribution of this study should be clearly clarified in abstract and introduction.

2.       Please broaden and update literature review to demonstrate the excellent performance of CNN or deep learning in resolving real problems. E.g., Torsional capacity evaluation of RC beams using an improved bird swarm algorithm optimised 2D convolutional neural network; Automated damage diagnosis of concrete jack arch beam using optimized deep stacked autoencoders and multi-sensor fusion.

3.       The performance of deep learning model is heavily dependent on the setting of hyperparamters. How did the authors tune/optimise the network hyperparameters to achieve the best classification performance in this research?

4.       Please add a photo to demonstrate the data (infrared spectroscopy)

5.       How about robustness of proposed method against noise effect?

6.       A comparison with other methods in literature is suggested to demonstrate the superiority of proposed method.

7.       More future research should be included in conclusion part.

Acceptable

Author Response

(The authors gave the same response as above.)

Round 2

Reviewer 1 Report

The authors improved the quality of the manuscript through supplementary data for reviewer's comments, but still some were not resolved.

Additional comments for above are as follows.

1. The authors mentioned that the performance was improved by modifying the existing CNN model from 2-D to 1-D layer and adding a BN layer.

    If so, the authors shall compare the performance of the traditional(unmodified) CNN and the proposed (modified) CNN, not the mere performance comparison with basic models such as SVM and NB.

2. Regarding CNN-LSTM combination, the authors shall show performance comparison and analysis before/after tuning. (similiar comments #1)

3. Regarding the authors' answer of applicability to the existing dataset,

   rather than relying on a specific dataset, the authors shall show various applicability.

   That is, unfortunately, it is difficult to evaluate the quaility of this manuscript highly if it is dependent only on a specific dataset.

4. Authors shall show the real picture of the mentioned Vis-SWIR equipment and a topology for the experiment.

5. The authors mentioned that actual photos(images) of the food were not gathered.

 Unfortunately, the readers will find it difficult to understand the experimental process because they cannot know the condition of the target food.

   This fact makes it difficult to evaluate the quality of this manuscript highly.

Author Response

(The authors gave the same response as above.)

Reviewer 2 Report

Comment 1

Line 6: Format Line 6 such as Line 7

Comment 2

Lines 146 - 147: 

spectra. [30] (Available from: 

http://dx.doi.org/10.17632/yhyzmp8rtb.2)

The authors should replace

spectra [30] (Available from: 

http://dx.doi.org/10.17632/yhyzmp8rtb.2).

Comment 3

Line 195

The authors must check if the work "Subsubsection" is right.

Comment 4

Lines 266 - 267

It's not so good to start the sub-section at the bottom of the page without text

Comment 5

Lines 359 - 361, Lines 371 - 377, 379 - 382

The authors should delete the Lines.

Comment 6

Lines 455 - 456

et al. "Identification

The authors should delete the ". 

Comment 7

References

The authors must format the References according to the journal's instructions

References should be described as follows, depending on the type of work:

Journal Articles:

1. Author 1, A.B.; Author 2, C.D. Title of the article. Abbreviated Journal Name Year, Volume, page range.

Author Response

(The authors gave the same response as above.)

Round 3

Reviewer 1 Report

The authors highly improved the quality of manuscript by reflecting reviewers' comments.

Here are just a minor comments:

Because the resolution of Figures 1 and 5 is low, it is suggested to replace them with vector type images.

Author Response

(The authors gave the same response as above.)
